# Jowiseungchungtang Inhibits Amyloid-β Aggregation and Amyloid-β-Mediated Pathology in 5XFAD Mice

**DOI:** 10.3390/ijms19124026

**Published:** 2018-12-13

**Authors:** Soo Jung Shin, Yu-on Jeong, Seong Gak Jeon, Sujin Kim, Seong-kyung Lee, Yunkwon Nam, Yong Ho Park, Dabi Kim, Youn Seok Lee, Hong Seok Choi, Jin-il Kim, Jwa-Jin Kim, Minho Moon

**Affiliations:** 1Department of Biochemistry, College of Medicine, Konyang University, 158, Gwanjeodong-ro, Seo-gu, Daejeon 35365, Korea; tlstnzz83@gmail.com (S.J.S.); yuon918@naver.com (Y.-o.J.); jsg7394@naver.com (S.G.J.); aktnfl3371@naver.com (S.K.); tjdrud7087@gmail.com (S.-k.L.); yunkwonnam@gmail.com (Y.N.); znf900809@naver.com (Y.H.P.); ltszzang123@naver.com (Y.S.L.); redhill97@gmail.com (H.S.C.); 2Department of Medical Science, School of Medicine, Chungnam National University, Daejeon 35015, Korea; db941012@naver.com; 3Department of Nursing, College of Nursing, Jeju National University, Jeju-si 63243, Korea; 4Department of Biomedical Science, Jungwon University, Geosan, Chungbuk 28024, Korea; 5Department of Nephrology, School of Medicine, Chungnam National University, Daejeon 35015, Korea

**Keywords:** Alzheimer’s disease, amyloid-β, jowiseungchungtang, 5XFAD mice, neurodegeneration, neuroinflammation, adult hippocampal neurogenesis

## Abstract

Alzheimer’s disease (AD) is a neurodegenerative disease, which is accompanied by memory loss and cognitive dysfunction. Although a number of trials to treat AD are in progress, there are no drugs available that inhibit the progression of AD. As the aggregation of amyloid-β (Aβ) peptides in the brain is considered to be the major pathology of AD, inhibition of Aβ aggregation could be an effective strategy for AD treatment. Jowiseungchungtang (JWS) is a traditional oriental herbal formulation that has been shown to improve cognitive function in patients or animal models with dementia. However, there are no reports examining the effects of JWS on Aβ aggregation. Thus, we investigated whether JWS could protect against both Aβ aggregates and Aβ-mediated pathology such as neuroinflammation, neurodegeneration, and impaired adult neurogenesis in 5 five familial Alzheimer’s disease mutations (5XFAD) mice, an animal model for AD. In an in vitro thioflavin T assay, JWS showed a remarkable anti-Aβ aggregation effect. Histochemical analysis indicated that JWS had inhibitory effects on Aβ aggregation, Aβ-induced pathologies, and improved adult hippocampal neurogenesis in vivo. Taken together, these results suggest the therapeutic possibility of JWS for AD targeting Aβ aggregation, Aβ-mediated neurodegeneration, and impaired adult hippocampal neurogenesis.

## 1. Introduction

Alzheimer’s disease (AD) is the most common aging-associated neurodegenerative diseases. The hallmark symptoms of AD include memory deficits and cognitive dysfunction. The major lesions of AD are amyloid-β (Aβ) plaques, the aggregated form of Aβ, and neurofibrillary tangles which are composed of abnormally phosphorylated tau proteins [1,2]. The amyloid hypothesis proposes that the aggregation of Aβ triggers AD pathologies such as synaptic dysfunction, neuroinflammation, and neurodegeneration [3].

The Aβ peptide is derived from proteolysis of the amyloid precursor protein (APP) [4]. Proteolysis consecutively progresses through a series of β-secretase and γ-secretase processing steps [5]. When the activities of these enzymes are disrupted or sites at which the enzymes cleave APP are blocked, Aβ aggregation and accumulation are increased. In addition, mutations in the *APP* gene can change the sequence of amino acids at the cleavage sites of the β-secretase and presenilin 1 (*PSEN1*)/*PSEN2* mutation alters γ-secretase-dependent functions [6]. These abnormal cleavage processes result in the production of insoluble Aβ peptides and formation of oligomer aggregates [7]. Unless the aggregated Aβ peptides are cleared, Aβ accumulates in the brain [8]. These neurotoxic Aβ aggregates lead to hallmark symptoms of AD such as synapse loss, neuroinflammation, memory impairment, and cognitive dysfunction [9,10]. The Aβ aggregation and accumulation in aging brains causing the AD pathogenesis is known as the amyloid hypothesis [11,12]. No substantial disease-modifying treatment has been developed and approved to date. Instead, symptom-treating drugs such as cholinesterase inhibitors and *N*-methyl d-aspartate (NMDA) receptor antagonists have approved [13]. Many studies support the detrimental effects of Aβ accumulation on neuronal functions [14,15]. Recently, a study indicated that Aβ oligomer-induced neurotoxicity may be reversible, suggesting a potential direction of research to develop a treatment for AD [16]. Therefore, inhibiting Aβ aggregation might still be considered as the most promising treatment strategy for AD.

Adult hippocampal neurogenesis, the generation of new neurons in the subgranular zone of the dentate gyrus (DG) in the hippocampus, has been known to be responsible for maintenance of learning and memory [17]. Previous studies have reported that altered levels of adult hippocampal neurogenesis in patients and animals with neurodegenerative diseases result in cognitive impairments as seen in Parkinson’s disease [18,19]. Although there are inconsistent findings regarding the level of changes in proliferation and differentiation in the hippocampus of AD patients and animal models [20], there is a general consensus that altered adult hippocampal neurogenesis is associated with cognitive dysfunction in AD. Therefore, regulating adult hippocampal neurogenesis could be a therapeutic strategy for AD treatment [21,22,23] and it is hypothesized that treatment for cognitive impairment in AD may be effective when it is incorporated into a strategy for increasing adult hippocampal neurogenesis.

Jowiseungchungtang (JWS) is an herbal formula consisting of 14 components. For a long time, JWS has been prescribed to treat dementia, obesity, oxidation, and depression in oriental medicine [24,25,26,27]. Previous studies have reported that JWS modulates oxidative stress, lipid metabolism, and inflammation responses [25]. Furthermore, there are studies demonstrating cognition-enhancing effects of JWS in rodents [28,29]. Although no studies have investigated the effects of JWS on adult hippocampal neurogenesis, one previous study reported that extracts of the Platycodi Radix, one of the components of JWS, increases adult hippocampal neurogenesis [30]. Considering that traditional oriental medicines could attenuate cognitive impairments and AD-related symptoms [31], JSW-induced cognitive improvement could be useful in clinical practice.

However, specific mechanisms including histochemical changes underlying the cognition-enhancing effects of JWS in AD patients have not been examined. In particular, no studies have been conducted regarding the effects of JWS on Aβ aggregation, Aβ-induced neuropathologies, and adult hippocampal neurogenesis. These prompted us to examine whether JWS affects Aβ aggregation, Aβ-mediated neuropathology, and adult hippocampal neurogenesis in 5 five familial Alzheimer’s disease mutations (5XFAD) mice, a mouse model of AD. Here, we first report histologic evidence that JWS can ameliorate Aβ_42_ aggregation, neuroinflammation, and neurodegeneration in the neocortex, and increase adult hippocampal neurogenesis in 5XFAD mice.

## 2. Results

### 2.1. Inhibitory Effects of JWS on Aβ Aggregation In Vitro

To investigate the effect of JWS on Aβ_42_ aggregates in vitro, we performed the Thioflavin T (ThT) assay. ThT is a widely used cationic benzothiazole dye to identify amyloid fibrils since it preferentially binds to channels formed by aromatic or hydrophobic residues on the surface of β-sheets. In addition, its fluorescence enhancement property upon binding to amyloid fibrils contributes to the visualization of Aβ aggregation via changes in the fluorescence intensity [32,33]. Vehicle (100%) and JWS (80% at 400 μg/mL; 60% at 600 μg/mL) were compared to a positive control group (Morin, 40% compared to vehicle). Aβ_42_ aggregation was significantly decreased with JWS at 600 µg/mL (Figure 1). The aggregates of monomeric Aβ_42_ were inhibited by JWS treatment in a dose-dependent manner, as shown by reduced ThT fluorescence intensity in the presence of JWS.

### 2.2. Inhibitory Effects of JWS on Aβ Accumulation in the Cerebral Cortex of 5XFAD Mice

In addition to the ThT assay, immunohistochemical stainings were performed to confirm the anti-Aβ aggregation effect in vivo. Using 4G8 antibodies, we examined the inhibitory effect of JWS on Aβ accumulation in the brain of 5XFAD mice (Figure 2A). Compared with vehicle-administered 5XFAD mice, JWS-administered 5XFAD mice displayed fewer 4G8-positive areas in the neocortex (Figure 2B). To further investigate whether JWS can effectively inhibit Aβ aggregation, Aβ aggregates were labeled by ThS staining (Figure 2C). In comparison to 5XFAD mice + Vehicle (0.66%) and 5XFAD mice + JWS (0.3%), the amount of Aβ plaques in the deep cortical layers of 5XFAD mice + JWS was significantly reduced by 0.37% (Figure 2D). The result demonstrates that JWS reduces Aβ plaque in the brain of Aβ-overexpressing transgenic mice.

### 2.3. Inhibitory Effects of JWS on Neuroinflammation in the Cerebral Cortex of 5XFAD Mice

Neuroinflammation induced by neurotoxic Aβ aggregates such as Aβ oligomers recruits and activates microglia and astrocytes, and thereby exacerbates the pathophysiological prognosis of AD with symptoms of gliosis, neurodegeneration, and cognitive impairment [34,35,36]. To investigate whether the reduced accumulation of Aβ due to the administration of JWS attenuates inflammation, we performed immunohistochemical staining using ionized calcium-binding adapter molecule 1 (Iba1), a marker of microglial cells and GFAP, a marker of astrocytes. As a result of immunological activity in the deep cortical layers of the brain, not only was the number of microglial cells increased in 5XFAD mice compared to wild-type (WT) mice, but morphological activation was also increased (Figure 3A). Quantification of the immunoreactivity revealed a significant increase in the Iba1-positive area in 5XFAD mice (4.22%) compared with WT mice (1.48%). However, in JWS-treated 5XFAD mice (3.20%), a significantly reduced Iba1-positive area was observed compared to the vehicle-treated 5XFAD mice (Figure 3B). Similarly, while the glial fibrillary acidic protein (GFAP)-positive area in vehicle-treated 5XFAD mice was significantly increased compared to WT mice (0.26%), the GFAP-positive area was significantly reduced in JWS-treated 5XFAD mice (4.50%) compared to vehicle-treated 5XFAD mice (7.37%) (Figure 3C,D). These results demonstrate that administration of JWS can reduce the increase in the number of microglial cells and astrocytes in Aβ accumulation-induced neuroinflammation in 5XFAD mice.

### 2.4. Inhibitory Effects of JWS on Neurodegeneration in the Cerebral Cortex of 5XFAD Mice

The loss of neurons and synapses is a major factor directly correlated with cognitive impairment and is well known to occur in neurodegenerative diseases such as AD [37,38,39]. As JWS reduced both Aβ accumulation and neuroinflammation in 5XFAD mice, we investigated the effect of JWS on neuronal cell death and synaptic loss in deep cortical layers (Figure 4A). Immunohistochemical stainings with NeuN (neuronal nuclei), which is a marker of the mature neuronal nucleus, showed that the number of NeuN-positive cells per area is significantly reduced in vehicle-treated 5XFAD mice compared to WT mice and was significantly restored in JWS-treated 5XFAD mice (Figure 4B). Likewise, we evaluated the protective effect of JWS on synaptic loss through immunohistochemical stainings using synaptophysin (SYN), a marker for presynaptic vesicles (Figure 4C). In vehicle-treated 5XFAD mice, the fluorescence intensity of SYN immunoreactivity was significantly reduced compared to WT mice. On the other hand, the fluorescence intensity of SYN was significantly restored in JWS-treated 5XFAD mice compared to vehicle-treated 5XFAD mice (Figure 4D). These results indicate that JWS has a neuroprotective effect that alleviates the loss of neurons and synapses mediated by Aβ aggregation in 5XFAD mice.

### 2.5. Inhibitory Effects of JWS on Impairment of Adult Hippocampal Neurogenesis in 5XFAD Mice

It is well established that the Aβ-induced deficit in adult hippocampal neurogenesis might contribute to cognitive impairment in AD patients [23]. Therefore, to investigate if JWS with Aβ aggregation-inhibiting activity may affect adult hippocampal neurogenesis, we performed immunofluorescence labeling using anti-Ki67 and anti-doublecortin (DCX) antibodies to detect the proliferation and differentiation of adult hippocampal neural stem cells. A quantitative analysis revealed that the number of Ki-67- and DCX-stained cells were reduced in the hippocampus of vehicle-administered 5XFAD mice compared to the vehicle-administered WT mice (Figure 5). In contrast, immunoreactivity of Ki-67 (Figure 5A,B) and DCX (Figure 5C,D) indicated that JWS-treated 5XFAD mice exhibited a significantly increased number of Ki-67- and DCX-positive cells compared to vehicle-treated 5XFAD mice. These results suggest that JWS effectively enhances adult hippocampal neurogenesis in 5XFAD mice.

## 3. Discussion

After the amyloid cascade hypothesis has been proposed and adopted for decades as the most evident theory to explain the causes of AD, a lot of clinical and non-clinical trials have been conducted to treat AD. Among the different strategies for treating AD, alternative medicines including oriental herbal medicines are paid considerable attention to due to their long history of clinical use [40]. Several studies have examined cognition-improving effects of herbal formulae using JWS, one of the commonly prescribed herbal medicines for dementia patients [28,41,42]. However, the precise pathological changes underlying the beneficial effects of JWS in AD have not yet been examined. In the present study, we demonstrate that JWS has inhibitory effects on AD pathologies, including Aβ accumulation, neuroinflammation, neurodegeneration, and impaired adult hippocampal neurogenesis in an animal model of AD.

Several studies have reported anti-Aβ aggregation effects of herbal formulae such as *Ukgansan (Yi gan San* in Chinese and *Yokukansan* in Japanese) [43], and *Hwanglyeonhaedoktang* (*Huanglian jiedu tang* in Chinese and *Orengedokuto* in Japan) [44]. Similar to the formulae have anti-aggregation, both in vitro result of ThT assay and in vivo results of 4G8, ThS staining showed that JWS reduced endogenous Aβ accumulation in the neocortex. These results suggested that JWS has anti-accumulating effects on Aβ in the early phase of Aβ pathology since the neocortex is known to be a region where Aβ aggregates in the early phase of Aβ deposition [45]. Several studies have reported inhibitory effects of each constituent or its bioactive compound of JWS such as polygalae radix, Schisandra chinensis Baillon, and Liriope platyphylla on Aβ levels. In the study examining effects of polygalae radix on Aβ levels, it was demonstrated that polygalae radix decreases Aβ_1–40_ secretion by induction of autophagy in SH-SY5Y (human bone marrow-derived neuroblastoma) cell line [46]. Moreover, lignans, the bioactive compounds from Schisandrae Fructus, reduced Aβ_1–42_ secretion in the hippocampus of mice [47]. Another constituent, the extract of red Liriope platyphylla which is made from Liriopis Tuber, decreases Aβ_1–42_ levels in the brain of Tg2576 mice [48]. These studies might explain that the above-mentioned constituents may be involved in the anti-Aβ accumulation effects of JWS. Future studies are needed to clarify the specific mechanisms of JWS on Aβ accumulation.

Current trends in AD management are focused on prevention or early intervention [49,50]. We examined the deposition of Aβ in the neocortex, which is known to be the first area where Aβ deposition can be observed [45]. Importantly, as the 5XFAD mice used in the present study were young mice aged four months, our results indicate that administration of JWS could ameliorate Aβ accumulation in the early phase of AD. In addition, 5XFAD mice showed Aβ deposition at two months and significant hippocampus-dependent memory impairment at six months of age [51,52]. Therefore, it could be hypothesized that JWS could be applicable for the treatment of AD targeting Aβ accumulation before the onset of cognitive impairments.

JWS reduced microgliosis and astrogliosis and thereby reduced neurodegeneration in the neocortex. Although, the extent of decrement of Iba1 and GFAP positive areas were not as much as WT mice, it significantly decreased Iba1- and GFAP-positive areas compared to vehicle-treated mice. Moreover, as indicated by the immunohistochemical results of NeuN and SYN stainings, JWS protected against the loss of neurons and synapses which is associated with Aβ pathology. Since the accumulation of Aβ is considered a key mechanism of neurodegeneration in AD [53], our results indicate that JWS could suppress the neurodegeneration in AD which is associated with JWS’ anti-Aβ accumulation effects. These results are corroborated by previous studies investigating the effects of the constituents of JWS or their bioactive compounds on Aβ-induced neurodegeneration. Polygalae Radix and Acori Graminei Rhizoma, both constituents of JWS, have been shown to improve learning and memory performance by ameliorating Aβ-induced apoptotic cell death in the hippocampus of mice [54]. Another study has reported that Acori Graminei Rhizoma improves cognitive function by decreasing the activity of nitric oxide synthase in the cerebrum and the hippocampus [55]. Lignans from Schisandrae Fructus attenuated neurodegeneration by decreasing Aβ_1–42_ in the hippocampus of mice [47]. Therefore, the above-mentioned constituents may be involved in attenuating effect of JWS on Aβ-induced neurodegeneration.

Cognitive impairment is the most remarkable symptom in AD [56,57]. Reportedly, impaired cognitive functions including spatial learning and memory are correlated with decreased adult hippocampal neurogenesis in 5XFAD mice [58,59]. Immunohistochemical analysis of Ki67 and DCX, which are markers of cellular proliferation and neuronal differentiation, have shown that JWS could up-regulate adult hippocampal neurogenesis in the DG of the hippocampus. It was reported that extract of Platycodi Radix increases adult hippocampal neurogenesis in middle-aged mice [30]. α-asarone, a bioactive compound of Acori Graminei Rhizoma, enhances proliferation and differentiation in neural progenitor cells and the DG of cultured hippocampal slices [60]. Moreover, it was reported that schisandrin, a lignan extracted from Schisandrae Fructus, enhances dendritic outgrowth and synaptogenesis in cultured hippocampal neurons [61] and schisandrin B increases neurogenesis in the subventricular zone in mice [62]. Additionally, although levels of hippocampal neurogenesis were not affected, Polygalae Radix increases mRNA expression of glial cell line-derived neurotrophic factor and dendritic spine maturation in the hippocampus and the nucleus accumbens in mice [63]. These studies provide a possible explanation about specific ingredients that are involved in the up-regulation of adult hippocampal neurogenesis induced by JWS. In this context, further studies are needed to examine the effects of each constituent or its bioactive ingredients on Aβ-mediated neuropathology as well as pharmacological interactions among them.

In the present study, we demonstrated that JWS inhibits Aβ aggregation and accumulation, attenuates neuroinflammation and neuronal death, and restores adult hippocampal neurogenesis. To date, a number of trials examining the efficacy of a single-target drug on AD have failed since its multifactorial nature of pathogenesis. Therefore, AD treatment is changing to multi-target approaches [64,65].

A test for oral toxicity of JWS in mice based on the recommendation of Korea Food and Drug Administration guideline has reported that JWS can be considered as a safe drug [66]. Given the protective effects on neurodegeneration by inhibition of Aβ accumulation and improving effects on adult hippocampal neurogenesis of JWS, it can be speculated that JWS is capable to delay or conserve the cognitive performance of patients with early stages of AD.

Taken together, we examined the inhibitory effects on neurodegeneration via reduction of Aβ aggregation and accumulation in the neocortex and enhancing effects on adult hippocampal neurogenesis of JWS. The results indicate that JWS may alleviate the early stages of AD targeting Aβ aggregation and adult hippocampal neurogenesis.

## 4. Materials and Methods

### 4.1. Animals and JWS Administration

Animals were treated in accordance with the National Institutes of Health Guide for the Care and Use of Laboratory Animals (NIH Publications No. 8023, revised 1978) and under the supervision of the Institutional Animal Care and Use Committee at Konyang University (project code:P-16-02-A-01; 2 March 2016). The Aβ-overexpressing transgenic 5XFAD mice have mutations in the *APP* (Swe^K670N/M671L^, Lon^V717I^, and Flo^I716V^) and *PSEN1* (M146L and L286V) genes regulated by the Thy1 promoter. All animals were females and four months old. JWS was orally administered to littermate wild-type (WT) or 5XFAD mice once every two days for one month. The mice were divided into three groups (*n* = 5 in each group): (1) saline-administered WT mice, (2) saline-administered 5XFAD mice, and (3) JWS (2 g/kg)-administered 5XFAD mice. We obtained JWS from Kyung Hee University Korean Medicine Hospital (Gangdong-gu, Seoul, Korea). The components of JWS are as follows: Coicis Semen, Castaneae Semen, Raphani Semen, Longanae Arillus, Liriopis Tuber, Platycodi Radix, Acori Graminei Rhizoma, Thujae Semen, Zizyphi Semen, Massa Medicata Fermentata, Ephedrae Herba, Schisandrae Fructus, Amomi Semen, and Polygalae Radix [41] (Table 1).

### 4.2. Brain Tissue Preparation

Mice were anesthetized and perfused with 0.05 M PBS (phosphate buffered saline) before being post-fixed with cold 0.1 M phosphate buffer containing 4% paraformaldehyde (PFA). Next, the extracted brain tissue was fixed in 0.1 M phosphate buffer containing 4% PFA overnight at 4 °C before being stored in a cryopreservation solution made of 0.05 M PBS containing 30% sucrose. Serial coronal sections (30 μm thick) were prepared using a cryostat CM1850 microtome (Leica Biosystems, Wetzlar, Germany). Sections were stored at 4 °C in a cryoprotectant solution (25% ethylene glycol, 25% glycerol, and 0.05 M phosphate buffer) until the immunohistochemistry analysis.

### 4.3. ThT Assay

We used the ThT β-Amyloid (1–42) Aggregation Kit (ANASPEC; Fremont, CA, USA) for analysis of Aβ accumulation. We diluted ThT working solution from 20 mM to 2 mM by adding assay buffer to the ThT working solution. Then, we added cold assay buffer (1 mL) to 0.25 mg Aβ_42_ to obtain a 0.25 mg/mL Aβ_42_ peptide solution. Inhibitor solution was diluted to 2 mM stock solution Morin (positive control) from 20 mM with assay buffer. The Tecan Infinite M200 Fluorescence Microplate Reader (Tecan, Männendorf, Switzerland) was used to detect the intensity of ThT fluorescence at wavelengths of 355–450 nm.

### 4.4. Thioflavin S (ThS) Staining

After the sectioned mouse brains were washed three times for five minutes in PBS, the brains were incubated with ThS solution (50 mg of 5 mg/mL of ThS (Sigma-Aldrich, St. Louis, MO, USA), 5 mL of distilled water and 5 mL of EtOH (99%)) at room temperature for ten minutes. After that, the tissues were incubated with 50% ethanol two times for three minutes and subsequently washed with PBS three times for three minutes. Finally, the stained brain sections were mounted on glass slides with Richard–Allan Scientific mounting medium (Thermo scientific, Waltham, MA, USA).

### 4.5. Immunofluorescence Labeling and Quantification

The brain sections were washed three times for five min each in PBS and then incubated with primary antibodies: a mouse anti-Aβ_17–24_ antibody (4G8; 1:2000 dilution, Biolegend, San Diego, CA, USA), a goat anti-ionized calcium-binding adapter molecule 1 antibody (Iba1; 1:500 dilution, Abcam, Cambridge, UK), a rat anti-glial fibrillary acidic protein antibody (GFAP; 1:200 dilution, Thermo Fisher, Waltham, MA, USA), a mouse anti-neuronal nuclei antibody (NeuN; 1:100 dilution, Merck Millipore, Burlington, MA, USA), a mouse anti-SYN antibody (1:500 dilution, Sigma-Aldrich, St. Louis, MO, USA), a goat anti-DCX antibody (1:50 dilution, Santa Cruz Biotechnology, Dallas, TX, USA), and a rabbit anti-Ki-67 antibody (1:200 dilution, Abcam, Cambridge, UK). Before immunostaining with 4G8 antibody, brain tissues were incubated with 70% formic acid for 20 min to retrieve the antigen. After incubation with primary antibodies overnight at 4 °C, the brain tissues were washed three times with PBS for five minutes each. The secondary antibodies (1:200 dilution) were applied at room temperature for 1.5 h. The following secondary antibodies were used: Alexa Fluor 488 goat anti-mouse IgG, Alexa Fluor 488 donkey anti-goat IgG, Alexa Fluor 488 goat anti-rat IgG, Alexa Fluor 488 donkey anti-rabbit IgG, and Alexa Fluor 594 donkey anti-goat IgG. Brain slices were washed three times for five minutes in PBS solution. The immuno-labeled tissues were mounted on glass slides with Fluoroshield mounting medium with DAPI (Sigma-Aldrich, St. Louis, MO, USA). Fluorescence was recorded with a Carl Zeiss LSM 700 Meta confocal microscope (Carl Zeiss, Oberkochen, Germany). Analysis and quantification of immunoreactivity were performed by Image J software (1.50i version, National Institutes of Health, Sacaton, AZ, USA).

### 4.6. Statistical Analysis

All histological analyzes were performed randomly with a blind manner for each group. All statistical parameters were calculated using GraphPad Prism 5.0 software (GraphPad Software, San Diego, CA, USA). Values were expressed as mean ± standard deviation (SD). Data were analyzed using the Student’s *t*-test for comparisons between the two groups and the one-way analysis of variance (ANOVA) followed by the Tukey’s post hoc test for comparisons between three groups. Differences with a *p*-value smaller than 0.05 were considered statistically significant.

## 5. Conclusions

The present study demonstrates that JWS could ameliorate Aβ_42_ aggregation as well as AD pathologies including neuroinflammation and neurodegeneration in the neocortex of 5XFAD mice. In addition, JWS also attenuates impaired adult hippocampal neurogenesis of 5XFAD mice (Figure 6). Therefore, it can be concluded that JWS might have protective effects on Aβ-mediated neuropathology in AD.

## Figures and Tables

**Figure 1 ijms-19-04026-f001:**
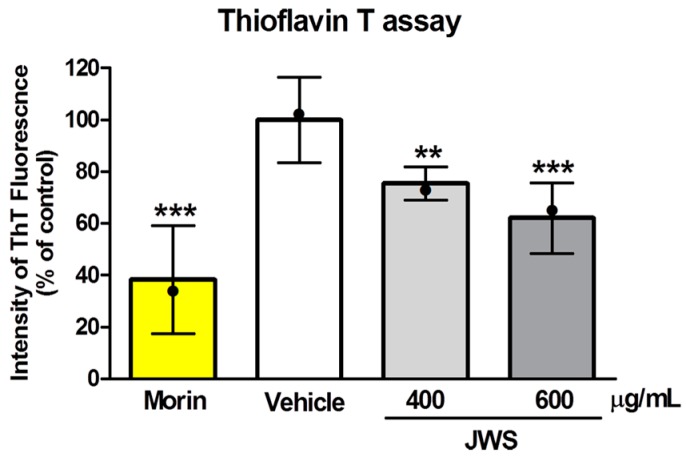
Fluorescence emission intensity of Thioflavin T in the presence or absence of Jowiseungchungtang (JWS). The extent of aggregation of monomeric Aβ_42_ peptides after incubation with morin, vehicle, 400 μg/mL of jowiseungchungtang (JWS), or 600 μg/mL of JWS were examined using the ThT assay kit. Data are presented as mean ± SD. Significance by Tukey’s post hoc test following one-way analysis of variance (ANOVA) is indicated as *** *p* < 0.001 and ** *p* < 0.01 versus the vehicle-treated group. Median value is indicated as •.

**Figure 2 ijms-19-04026-f002:**
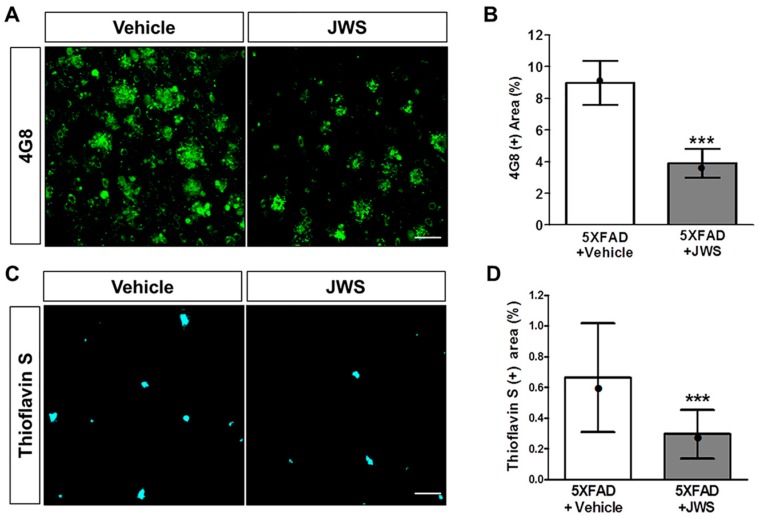
Histochemical evidence for decreased Aβ accumulation in the deep cortical layers in the brain of 5 five familial Alzheimer’s disease mutations (5XFAD) mice. Representative images of brain sections immunostained with anti-4G8 antibody (**A**) and stained with thioflavin S (ThS) (**C**). Scale bar = 100 μm. Jowiseungchungtang (JWS) significantly reduced the 4G8 (+) area (**B**) and ThS (+) area (**D**). Data are presented as mean ± SD (*n* = 5 in each group). Significance by Student’s *t*-test is indicated as *** *p* < 0.001 versus 5XFAD + vehicle group. Median value is indicated as •.

**Figure 3 ijms-19-04026-f003:**
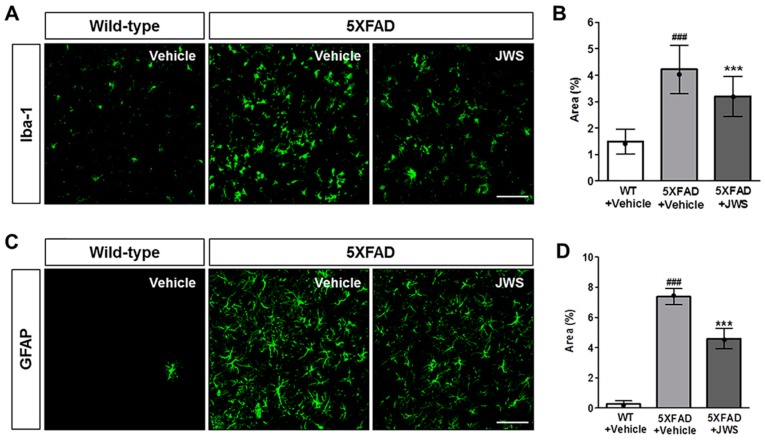
Immunofluorescence evidence of decreased neuroinflammation in the deep cortical layers of the brain in 5XFAD mice. Representative images of brain sections immunostained with anti-ionized calcium-binding adapter molecule 1 (Iba1) antibody (**A**) and anti-glial fibrillary acidic protein (GFAP) antibody (**C**). Scale bar = 100 μm. Jowiseungchungtang (JWS) significantly reduced the Iba1 (+) area (**B**) and GFAP (+) area (**D**). Data are presented as mean ± SD (*n* = 5 in each group). Significance by Tukey’s post hoc test following one-way ANOVA is indicated as *** *p* < 0.001 versus 5XFAD + vehicle group and ^###^
*p* < 0.001 versus WT + vehicle group. Median value is indicated as •.

**Figure 4 ijms-19-04026-f004:**
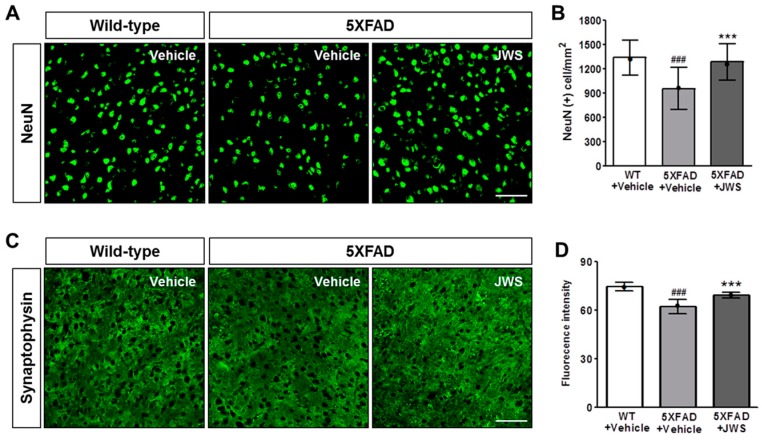
Immunofluorescence evidence of decreased neurodegeneration in the deep cortical layers of the brain in 5XFAD mice. Representative images of brain sections immunostained with anti-NeuN antibody (**A**) and anti-synaptophysin (SYN) antibody (**C**). Scale bar = 100 μm. Jowiseungchungtang (JWS) significantly reduced the NeuN (+) area (**B**) and SYN (+) area (**D**). Data are presented as mean ± SD (*n* = 5 in each group). Significance by Tukey’s post hoc test following one-way ANOVA is indicated as *** *p* < 0.001 versus 5XFAD + vehicle group and ^###^
*p* < 0.001 versus WT + vehicle group. Median value is indicated as •.

**Figure 5 ijms-19-04026-f005:**
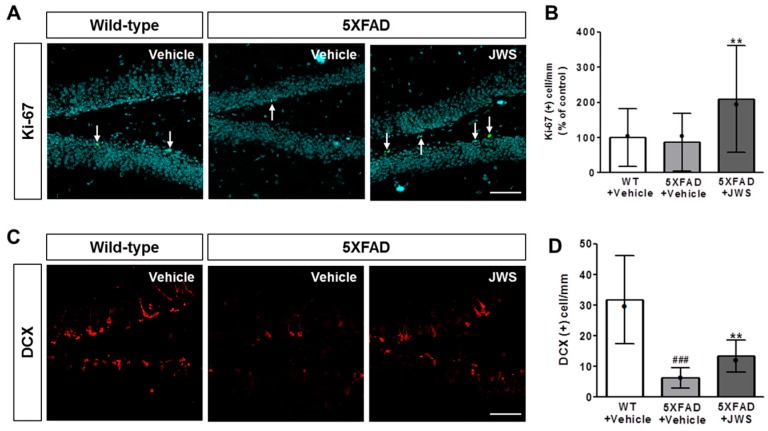
Effects JWS on cell proliferation and differentiation. (**A**) Representative images of brain sections immunostained with anti-Ki67 antibody. (**B**) The numbers of Ki67-positive cells in mice in the jowiseungchungtang (JWS)-treated group is higher compared to mice in the 5XFAD vehicle group. (**C**) Representative images of brain sections immunostained with anti-DCX antibody. (**D**) The numbers of DCX-positive cells in mice in the JWS-treated group was higher compared to mice in the 5XFAD vehicle group. Scale bar = 100 μm. Data are presented as mean ± SD (*n* = 5 in each group). Significance by Tukey’s post hoc test following one-way ANOVA is indicated as ** *p* < 0.001 versus 5XFAD + vehicle group and ^###^
*p* < 0.001 versus WT + vehicle group. Median value is indicated as •.

**Figure 6 ijms-19-04026-f006:**
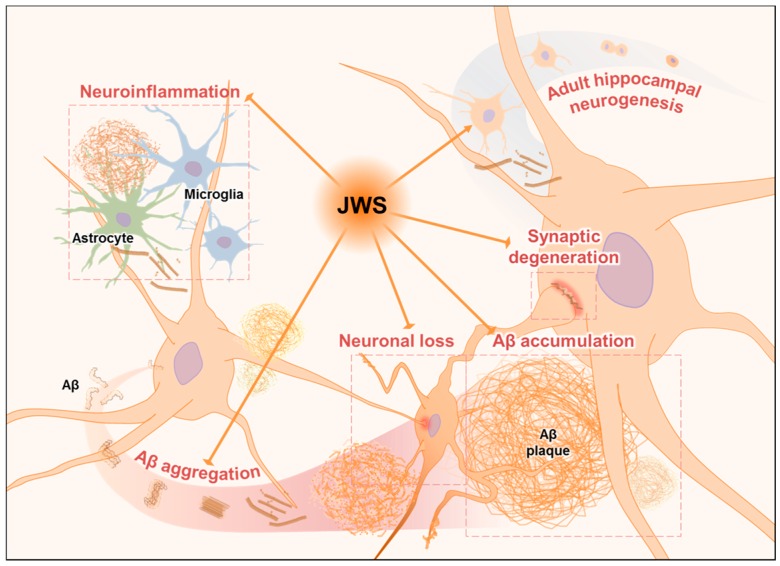
A schematic diagram of the protective effect of jowiseungchungtang (JWS) on Aβ-induced pathology. It is possible that JWS up-regulates adult hippocampal neurogenesis and inhibits Aβ accumulation and thereby decreases neuroinflammation and neurodegeneration in 5XFAD mice. T bar indicates suppression, and arrow indicates enhancement.

**Table 1 ijms-19-04026-t001:** The composition of jowiseungchungtang.

Constituents	Amount
Coicis Semen	8.0 g
Castaneae Semen	8.0 g
Raphani Semen	6.0 g
Longanae Arillus	6.0 g
Liriopis Tuber	4.0 g
Platycodi Radix	4.0 g
Acori Gramineri Rhizoma	4.0 g
Thujae Semen	4.0 g
Zizyphi Semen	4.0 g
Massa Medicata Fermentata	4.0 g
Ephedrae Herba	3.0 g
Schisandrae Fructus	3.0 g
Amomi Semen	3.0 g
Polygalae Radix	3.0 g
Total amount	64.0 g

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
