# Peer review of "Jowiseungchungtang Inhibits Amyloid-β Aggregation and Amyloid-β-Mediated Pathology in 5XFAD Mice"

_ijms, 2018, doi:10.3390/ijms19124026_

Round 1
Reviewer 1 Report
In this study the authors examined the effects of JWS on different outcome parameters in an animal model of AD. They observed less A-beta aggregation/pathology and improved neurogenesis. Based on these data the authors conclude that JWS could be considered as a possible treatment for AD.
The authors refer to two studies in which the effects of JWS were tested in humans. I am not impressed by these studies because of a very low number of participants, and appropriate control condition (placebo). The authors should not refer to these studies as they may give a reader the impression that JWS has precognitive effects in humans and could then be biased.
I am missing a group: WT + JWS. This could have given information to what extend the effects of JWS are limited to AD or whether it also has an effect in a ‘healthy’ brain.
With respect to JWS, is there any data on the PK characteristics of JWS. The authors administered JWS every two days for one month. If the half-life of the compound is short then I wonder how these effects could be achieved.
The group size was n=5. Did the authors perform a power analysis? This should have been done. Were the experimenters blind to the treatment conditions? This is quite essential for a valid and reliable experiment.
There is only a marginal attempt to discuss the possible mechanism of action of JWS and its active ingredients. The authors only report some general effects that are found in different systems.
The authors should caution their conclusion since there are many drugs that have shown beneficial effects in AD mice (also on cognition), but were not effective in patients.
Author Response
November 27, 2018
[Reviewer 1]
In this study the authors examined the effects of JWS on different outcome parameters in an animal model of AD. They observed less A-beta aggregation/pathology and improved neurogenesis. Based on these data the authors conclude that JWS could be considered as a possible treatment for AD.
[1] The authors refer to two studies in which the effects of JWS were tested in humans. I am not impressed by these studies because of a very low number of participants, and appropriate control condition (placebo). The authors should not refer to these studies as they may give a reader the impression that JWS has precognitive effects in humans and could then be biased.
Response: We fully agree with the suggestion of the reviewer. Following the advice of the reviewer, we excluded from the manuscript the citation of studies that could give the reader a biased impression.
[2] I am missing a group: WT + JWS. This could have given information to what extend the effects of JWS are limited to AD or whether it also has an effect in a ‘healthy’ brain.
Response: Thank you for the reviewer’s comment. As advised by the reviewer, the effect of JWS in a healthy brain is needed to determine whether the efficacy of JWS is limited to AD. However, our study focused on the effects of JWS on Aβ-mediated pathology. Therefore, the aim of our study is to determine examine whether JWS affects "Aβ" aggregation, Aβ-mediated neuropathology, and adult hippocampal neurogenesis in 5XFAD mice, a mouse model of AD. Although there is no literature that studies the efficacy of JWS in healthy brain, few studies have demonstrated the cognitive enhancing effects of JWS in rats [1, 2]. In addition, although it was a small group clinical study, cognitive function-enhancing effects were observed in normal elderly (n=10) as well as in early AD patients (n=15) when JWS was given for nine months [3]. Furthermore, as described in the manuscript, JWS also had a positive effect on rodent model of obesity and depression [4-7]. These results suggest that JWS may have beneficial effects on healthy brain, which is not limited to AD.
[3] With respect to JWS, is there any data on the PK characteristics of JWS. The authors administered JWS every two days for one month. If the half-life of the compound is short then I wonder how these effects could be achieved.
Response: Thank you for the comment. We agree with the concerns of the reviewer. Although there is no study that analyzes the PK characteristics of JWS, we suggest that based on previous studies demonstrating the efficacy of JWS, we may dilute concerns about the half-life of JWS. Not only was there a significant effect on the depression-rat model using JWS at similar dose (approximately 3.3g/kg) to our study [6], but also antioxidant and anti-inflammatory effects were found in obesity rat models using lower doses (100, 200 and 300 mg/kg) of JWS than our study [5]. In addition, JWS did not show significant toxicity up to 2,000 mg/kg that rodent maximum dose in the single-oral dose toxicity test [8]. Above all, the observed mitigation of Aβ-mediated neuropathology at our dose demonstrates the pharmacological effects of JWS.
[4] The group size was n=5. Did the authors perform a power analysis? This should have been done. Were the experimenters blind to the treatment conditions? This is quite essential for a valid and reliable experiment.
Response: We appreciate for this comment. With respect to power analysis, we performed in accordance with resource equation, which could be used when assuming effect size is difficult [9-11]. Based on resource equation, we considered E (error, acceptable range is 10 to 20) as 12. E could easily be measured by following formula: E = N (total number of animals) – T (number of groups). Thus, we calculated N as 15 (5 animals for each groups). Moreover, using this method could be justified by the fact that this might be suitable for 3Rs (Replacement, Reduction and Refinement) [12].
Regarding the blind in experiments, we believe that this non-clinical trial is not a case of considering placebo effects as in clinical trials, the administration of JWS proceeded non-blind. However, histological analysis was performed in a randomly with a blind manner for each group. We added this in methods section (line 336).
[5] There is only a marginal attempt to discuss the possible mechanism of action of JWS and its active ingredients. The authors only report some general effects that are found in different systems.
Response: Thanks for the comments. The comment of the reviewers is consistent with our limitations of this study. The JWS, consisting of 14 herbs have a lot of various bioactive components, and it has variety of effects on multiple targets. Moreover, the purpose of this study was to identify the effects of JWS as a standardized formula, not as separate ingredients. Thus, we were confirming only general effects of JWS. In addition, we also addressed some possible mechanisms of herbs such as Polygalae Radix, Schisandrae Fructus and Acori Graminei Rhizoma among the herbs that make up the JWS. As presented in the discussion, not only did the 14 herbs (and bioactivity components present in each herb) that make up the JWS individually have a therapeutic effect on Aβ-mediated neuropathology, but also might interact with each other to create complementary or synergistic therapeutic effects. We attempted to demonstrate the effectiveness of combination therapy on Aβ-mediated neuropathology by administration of known herbal formula JSW, to animal model of AD. The JWS significantly improved the Aβ-mediated neuropathologies compared to the control group. Nonetheless, we respect what the reviewer concern. Thus, we addressed the limitation of the study and suggestion regarding this issue in revised manuscript (line: 256-258).
[6] The authors should caution their conclusion since there are many drugs that have shown beneficial effects in AD mice (also on cognition), but were not effective in patients.
Response: We deeply sympathize with the comment of reviewer. As the reviewer pointed out, although a number of candidate therapeutic candidates for AD were effective in preclinical trials, they did not have a significant effect in clinical trials. Following the advice of the reviewer, we toned down the conclusions of the study (line: 32-33, 88, 271, 346-347).
References
1. Joo, Y. W.; Jong, W. K.; Whang, W. W.; Hyun, T. K.; Soon, K. P., An Experimental Study on the Effects of Jowiseungchungtang on Learning and Memory of Rats in the Radial - Arm Maze. Journal of oriental neuropsychiatry 1997, 8, (1), 69-79.
2. Wei-Wan, W.; Park Soon-Kwon; Lee Ung-Seok; Kim Hyun-Taek, The effects of Jowiseungchungtang on Learning and Memory of AD Rats using Morris water maze and Radial arm maze paradigm. Journal of KyungHee Oriental Medicine College 1998, 21, (1), 479-501.
3. Park, E. H.; Lee, Y. H.; Cho, S. H.; Kim, K. H.; Kim, B. G.; Kim, J. W.; W. W, H.; Kim, H. T. In Effects of Jowiseungchungtang on Cognitive Function and Auditory Event-related Potentials in Alzheimer disease, 2002; The Korean Psychological Association: 2002; pp 430-434.
4. Soon Kwon Park; Hong Jae Lee; Hyun Taek Kim; Wei Wan Whang, An experimental study of driental medicine on cure for dementia : the effect of Jowiseungcheongtang and Hyungbangjihwangtang on cure for aged rats. Journal of oriental neuropsychiatry 1998, 9, (2), 19-35.
5. Oh, S.-W.; Kim, B.-W., Effects of jowiseungcheung-tang extract on the lipid metabolism, anti-oxidation and inflammatory reflex high fat diet obese rats. The Journal of Internal Korean Medicine 2013, 34.
6. Ryu, J. M.; Kim, J. W.; Chi, S. E.; Kim, E. J.; Park, E. H.; Hwang, U. W., The effects of jowiseungchungtang versus fluoxetine in the chronic mild stress model of depression in rats. Journal of Oriental Neuropsychiatry 2004, 15, (1), 27-41.
7. Ansari, A.; Bose, S.; Yadav, M. K.; Wang, J. H.; Song, Y. K.; Ko, S. G.; Kim, H., CST, an Herbal Formula, Exerts Anti-Obesity Effects through Brain-Gut-Adipose Tissue Axis Modulation in High-Fat Diet Fed Mice. Molecules 2016, 21, (11).
8. Tae Young, J., Single Oral Dose Toxicity Test of Choweseuncheng-tang, a Polyherbal Formula in ICR Mice. Journal of Physiology & Pathology in Korean Medicin 2014, 28, (1).
9. Mead, R., The Design of Experiments: Statistical Principles for Practical Applications. Cambridge University Press: 1988.
10. Arifin, W. N.; Zahiruddin, W. M., Sample Size Calculation in Animal Studies Using Resource Equation Approach. The Malaysian journal of medical sciences : MJMS 2017, 24, (5), 101-105.
11. Charan, J.; Kantharia, N. D., How to calculate sample size in animal studies? Journal of pharmacology & pharmacotherapeutics 2013, 4, (4), 303-6.
12. Festing, M. F., How to reduce the number of animals used in research by improving experimental design and statistics. ANZCCART Fact Sheet 2011, 10, 1-11.

Reviewer 2 Report
This is a promising approach of AD´s treatment. This article demonstrated how JWS can overcome AD pathologies. This work is perfect in its current form. However, for additional evidence of the effects of JWS on AD, I would suggest to study the signaling pathways affected by JWS. Moreover, and taking into account that JWS is composed by 14 herbs, I would try each herb individually in order to know if one or two induce these effects.
Author Response
November 27, 2018
[Reviewer 2]
This is a promising approach of AD´s treatment. This article demonstrated how JWS can overcome AD pathologies. This work is perfect in its current form.
[1] However, for additional evidence of the effects of JWS on AD, I would suggest to study the signaling pathways affected by JWS. Moreover, and taking into account that JWS is composed by 14 herbs, I would try each herb individually in order to know if one or two induce these effects.
Response: We appreciate for the reviewer’s comments. We fully agree with what the reviewer interested. The JWS, consisting of 14 herbs have a lot of various bioactive components, and it has variety of effects on multiple targets. Moreover, the purpose of this study was to identify the effects of JWS as a standardized formula, not as separate ingredients. Thus, we were confirming only general effects of JWS. In addition, we also addressed some possible mechanisms of herbs such as Polygalae Radix, Schisandrae Fructus and Acori Graminei Rhizoma among the herbs that make up the JWS. As presented in the discussion, not only did the 14 herbs (and bioactivity components present in each herb) that make up the JWS individually have a therapeutic effect on Aβ-mediated neuropathology, but also might interact with each other to create complementary or synergistic therapeutic effects. We attempted to demonstrate the effectiveness of combination therapy on Aβ-mediated neuropathology by administration of known herbal formula JSW, to animal model of AD. The JWS significantly improved the Aβ-mediated neuropathologies compared to the control group. Nonetheless, we respect what the reviewer concern. Thus, we addressed the limitation of the study and suggestion regarding this issue in revised manuscript (line: 260-263).

Round 2
Reviewer 1 Report
I still think that the power analysis is not OK yet. There is a good tool to test how many subject you would need to conduct a good experiment. For example, the authors could use GPower (which can be downloaded). Based on an expected effect size and a predefined power you could get an estimate of the number of subjects needed.
In addition, I am very skeptical with respect to the standard errors in the graphs. It is hard to believe that this can be achieved with these methods with such a low group size. That's why I was asking about being blind to the treatment conditions.
Response: We have shown the median and range for our data and use the standard deviation (SD) not with the SEM for the values in our figures.